energy/computer modelling and simulation/fractals

coal oxygen adsorption, porous media, fractal, Sierpinski carpet, numerical simulation

**Authors for correspondence:**
Xiaoyu Liang
e-mail: xyliang@cjlu.edu.cn
Peng Xu
e-mail: xupeng@cjlu.edu.cn

# A numerical study on oxygen adsorption in porous media of coal rock based on fractal geometry

Xianzhe Lv[1], Xiaoyu Liang[1], Peng Xu[2,3] and Linya Chen[1]

[1]College of Metrology and Measurement Engineering and [2]College of Sciences, China Jiliang University, Hangzhou 310018, People's Republic of China
[3]State Key Laboratory Cultivation Base for Gas Geology and Gas Control, Henan Polytechnic University, Jiaozuo 454000, People's Republic of China

XLv, 0000-0002-0911-6021; XLi, 0000-0001-9871-1895; PX, 0000-0002-4349-5627

In order to explore the factors affecting coal spontaneous combustion, the fractal characteristics of coal samples are tested, and a pore-scale model for oxygen adsorption in coal porous media is developed based on self-similar fractal model. The liquid nitrogen adsorption experiments show that the coal samples indicate evident fractal scaling laws at both low-pressure and high-pressure sections, and the fractal dimensions, respectively, represent surface morphology and pore structure of coal rock. The pore-scale model has been validated by comparing with available experimental data and numerical simulation. The present numerical results indicate that the oxygen adsorption depends on both the pore structures and temperature of coal rock. The oxygen adsorption increases with increased porosity, fractal dimension and ratio of minimum to maximum pore sizes. The edge effect can be clearly seen near the cavity/pore, where the oxygen concentration is low. The correlation between the oxygen adsorption and temperature is found to obey Langmuir adsorption theory, and a new formula for oxygen adsorption and porosity is proposed. This study may help understanding the mechanisms of oxygen adsorption and accordingly provide guidelines to lower the risk of spontaneous combustion of coal.

# 1. Introduction

The spontaneous combustion of coal in the process of production, storage and transportation seriously threatens the safety of the coal industry. And the exploration of governing laws for the spontaneous combustion in coal is of practical significance for coal energy. A large amount of research results indicates that the

physical adsorption of coal to oxygen during spontaneous combustion is one of the causes of spontaneous combustion. Therefore, the study on coal oxygen adsorption can help to understand the mechanisms of coal spontaneous combustion [1–3].

Coal oxygen adsorption can reach its saturation in a very short time. However, the oxidation process is very slow. The premise of coal spontaneous combustion process is the physical adsorption of coal oxygen, which transports oxygen for subsequent chemical adsorption and reactions. A series of results show that the main factors affecting oxygen adsorption include temperature, particle size and moisture. Chen et al. [4] found that gas adsorption capacity of coal depends on water content, which is determined by the degree of coal deterioration. Liu et al. [5] discussed the effect of temperature on coal adsorption under low-pressure conditions through grand canonical Monte Carlo simulation and experiments. Qi et al. [6] stated that the oxygen consumption of coal is affected by the size of coal particles at a certain temperature. While Tang et al. [7] experimentally studied the effect of temperature on spontaneous combustion of coal and proposed the critical temperature for spontaneous combustion of different coal types. Several researchers found that gas adsorption in coal is related to the oxygen functional group by infrared spectroscopy, and the oxygen concentration affects the combustion of coal particles [8–11]. Recently, Xin et al. [12,13] carried out continuous measurement on sub-bituminous coal samples under different temperatures and oxygen concentrations, and analysed the chemical relationship between oxygen consumption and oxygen effect. They calculated four characteristic temperatures for the oxygen effect and the range of temperatures for the chemisorption using density functional theory and six different molecular models accordingly. Gao et al. [14] focused on the inhibiting effect of carbon dioxide on the process of coal oxygen adsorption by temperature-programmed experiment and indicated that 37–50% concentration of carbon dioxide can effectively prevent the spontaneous combustion of coal.

The mechanisms of coal spontaneous combustion are complex as physical and chemical interactions may occur at the same time, which are difficult to distinguish by experiments. Therefore, numerical simulation methods are employed to explore the spontaneous combustion mechanisms of coal. In order to characterize the pore-scale structures of coal, two kinds of geometrical models, experiment-based and computer-based methods, are generally used. For the experiment-based method, real geometric images of coal are directly obtained by scanning techniques such as computed tomography [15–18], magnetic resonance imaging [19,20], scanning electron microscopy [21] and synchrotron microscopy [22]. The computer-based method generates and characterizes a porous morphology according to a mathematical function. The flexibility and variability of digital reconstruction capabilities have advantages in design and research. Therefore, computational reconstruction methods are widely used in the literature. Some popular methods include defined particle swarms [23–25], Monte Carlo processes [26,27], discrete element methods [28], quadruple structure generation set methods [29,30] and other generation methods [31–35]. However, the multi-scale pore structure of coal is very complicated, and traditional methods are difficult to accurately and quantitatively characterize. Experimental data in various literature indicate that most natural porous media show fractal scaling characteristics. Therefore, fractal theory is applied to characterize the multi-scale structures of porous media and study the transport properties [36–38]. For example, Li et al. [39] determined the fractal dimension of shale by liquid nitrogen adsorption experiments and reported that the ranges of surface and pore fractal dimensions are 2.4227–2.6219 and 2.6049–2.7877, respectively. Kong et al. [40] studied the damage evolution mechanism of gas coal based on fractal theory.

Therefore, the liquid nitrogen adsorption experiments are carried out on coal samples to study the fractal characteristics of pore structures in the present work. And fractal geometry is employed to study the oxygen adsorption in coal porous media. This paper is organized as five sections. In §2, liquid nitrogen adsorption experiments are performed on six coal samples and the fractal dimensions are calculated accordingly. In §3, self-similar Sierpinski carpet model is used to characterize the pore structures of coal porous media, and the physical and mathematical models for transient oxygen adsorption are developed and validated. The effect of pore structural parameters and temperature on the oxygen adsorption is discussed in §4, and a quantitative relationship between oxygen adsorption and fractal dimension is also addressed. The concluding remarks are made in §5.

## 2. Fractal characteristics of coal

Coal porous media with complex and well-developed pore structures have strong adsorption capacity for oxygen. After oxygen molecules in the air enter the pores of the coal, physical adsorption of oxygen molecules occurs on pore surfaces under van der Waals force, and the physical composition of oxygen and coal does not change substantially during this process [41]. However, the complex pore structures

of coal porous media are difficult to characterize with traditional Euclidean geometry. Fortunately, fractal geometry indicates evident advantages in characterizing disorder and irregular objects and has made major breakthroughs in many research fields. A considerable number of studies have shown that natural porous media exhibit statistically self-similar fractal features, and the pore structures can be described by fractal theory [42–52]. Thus, liquid nitrogen adsorption experiments were carried out on six different coal samples to study the fractal characteristics of coal.

The fractal dimension of coal pore structures can be calculated using the Frenkel–Halsey–Hill (FHH) equation [53]

$$\ln V = A \ln \left[ \ln \left( \frac{P_0}{P} \right) \right] + B, \tag{2.1}$$

where $P$ and $P_0$, respectively, represent equilibrium and saturation pressure, $V$ is the volume of adsorbed gas and $A$ and $B$ are the slope and constant of the fitted line. Two fractal dimensions $D_1$ and $D_2$ were calculated by fitting the liquid nitrogen desorption curve shown in figure 1 at low-pressure and high-pressure sections, respectively. The fractal dimensions for six coal samples are summarized in table 1.

It can be seen from figure 1 that the liquid nitrogen desorption curve indicates clear fractal scaling laws in both low-pressure and high-pressure sections. In the low-pressure section, van der Waals force is the main adsorption force. And the gas adsorption capacity relates to the surface roughness of the coal sample. Thus, the fractal dimension $D_1$ for the low-pressure section represents the surface fractal dimension of the porous material. While the gas adsorption in the high-pressure section mainly depends on the capillary condensation, and the adsorption capacity is related to the pore structure of coal sample. That is the fractal dimension $D_2$ for high-pressure section represents the fractal dimension of pore structure. According to the concept of fractal, the value of fractal dimension in three-dimensional spaces is generally in the range from 2.0 to 3.0. It can be seen in table 1 that the fractal dimension of coal surface $D_1$ varies from 1.80 to 2.96. The determination of fractal dimension depends on the relative pressure $(P/P_0)$, which is empirically assigned to be 0.5. As shown in figure 1a, the critical pressure for surface and pore fractal dimensions is not evident. That is one of the main reasons that $D_1$ for coal sample #1 is smaller than 2.0. The fractal dimension for coal pore structure $D_2$ varies from 2.45 to 2.82. It should be noted that the pore fractal dimension in two-dimensional space equals the pore fractal dimension in three-dimensional space minus one if the area porosity is assumed to be the same as volume porosity [54,55]. Thus, the pore fractal dimension in two-dimensional space is in the range from 1.0 to 2.0.

# 3. Numerical simulation

It has been shown in §2 that the pore structures of coal samples indicate fractal scaling laws in both low-pressure and high-pressure sections. Therefore, the self-similar Sierpinski carpet model was used to simulate the microstructure of coal porous media and develop a pore-scale model for transient oxygen adsorption. As shown in figure 2, the coal structures can be generated by the Sierpinski carpet model. The side length $L_0$ and cut side length $C_0$ can be changed so that a porous medium model with different fractal dimensions and porosity can be generated with recursive algorithm. The fractal dimension of the Sierpinski carpet model can be determined by [56]

$$D_f = \frac{\ln (L_0^2 - C_0^2)}{\ln (L_0)}. \tag{3.1}$$

The porosity of the Sierpinski carpet model depends on both iteration methods and times [57]

$$\phi = \left( \frac{L_0^2 - C_0^2}{L_0^2} \right)^n, \tag{3.2}$$

where $n = 1, 2, 3 \ldots N$ is a positive integer and represents the order of the Sierpinski carpet model.

It can be found from equation (3.2) that the porosity depends on side length and cut side length as well as iteration order. Therefore, the actual coal sample can be simulated by changing the side length or truncating the side length and the fractal order. Due to computer capacity limitations, only five generations of the Sierpinski carpet model were simulated. For the two-dimensional Sierpinski carpet model, its fractal dimension is in the range of 1 to 2 [58,59]. In the following numerical simulation,

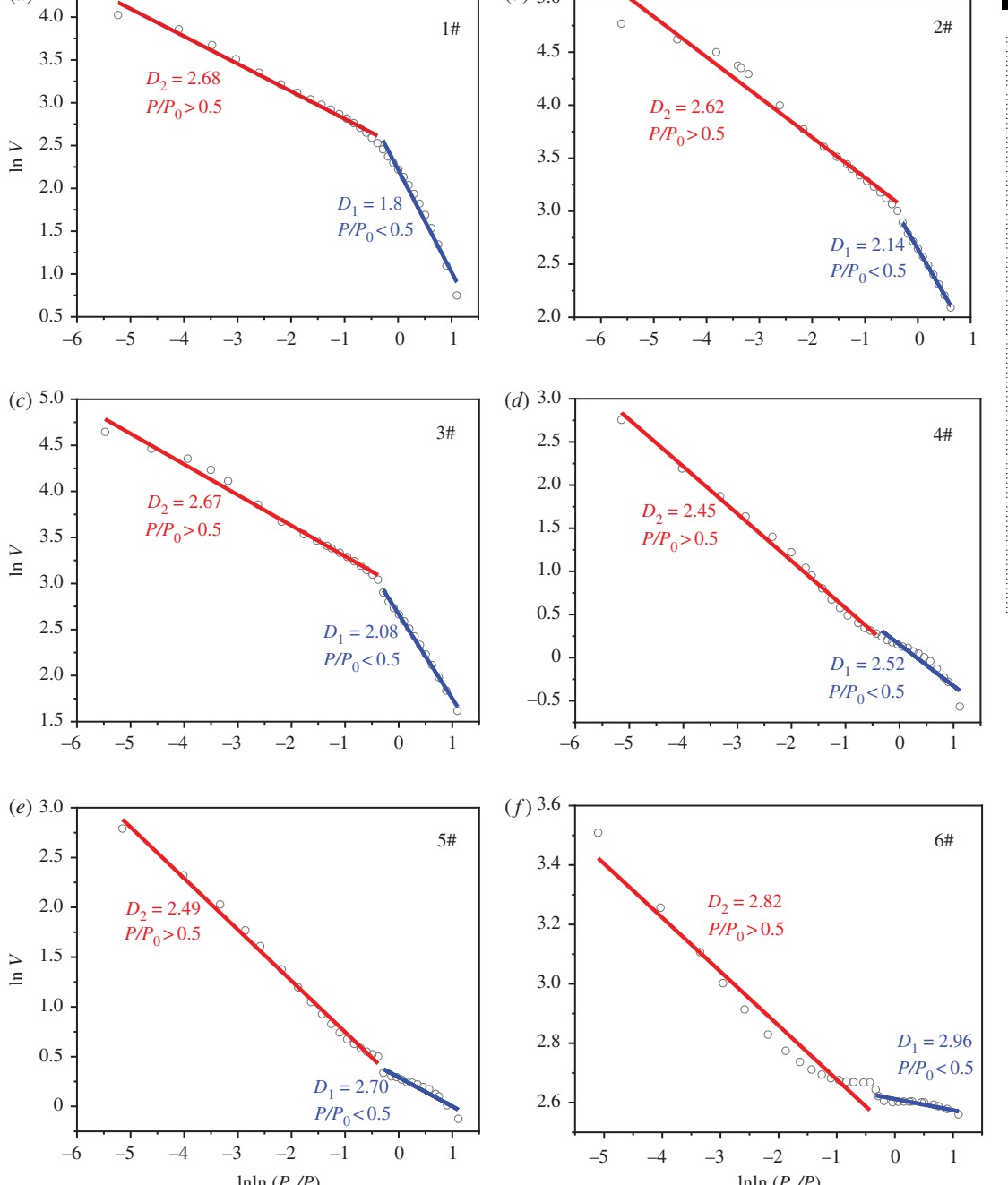

**Figure 1.** The liquid nitrogen desorption curve for six coal samples.

**Table 1.** Fractal dimensions of coal samples by liquid nitrogen adsorption experiment.

| coal sample | $A_1$ | $D_1 = 3 + A_1$ | $R^2$ | $A_2$ | $D_2 = 3 + A_2$ | $R^2$ |
|---|---|---|---|---|---|---|
| 1# | −1.20 | 1.80 | 0.98 | −0.32 | 2.68 | 0.98 |
| 2# | −0.86 | 2.14 | 0.99 | −0.38 | 2.62 | 0.97 |
| 3# | −0.92 | 2.08 | 0.99 | −0.33 | 2.67 | 0.99 |
| 4# | −0.48 | 2.52 | 0.90 | −0.55 | 2.45 | 0.99 |
| 5# | −0.30 | 2.70 | 0.90 | −0.51 | 2.49 | 0.99 |
| 6# | −0.04 | 2.96 | 0.76 | −0.18 | 2.82 | 0.95 |

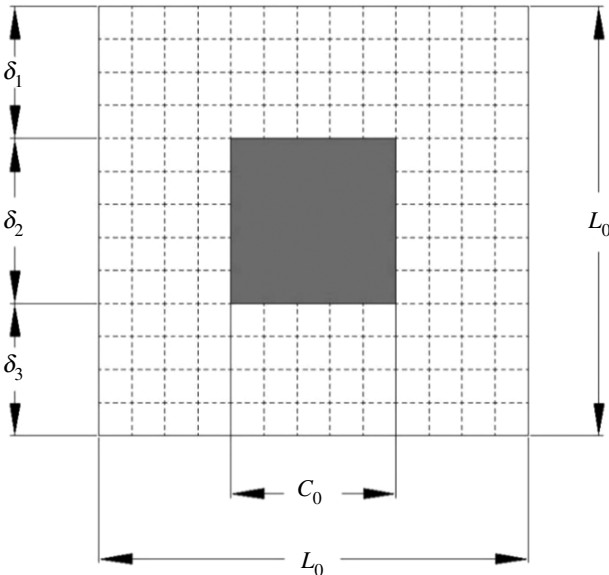

**Figure 2.** A schematic graph of the construction of the Sierpinski carpet model ($n = 1$, $L_0 = 13$, $C_0 = 5$).

**Table 2.** The parameters of the Sierpinski carpet models ($L_0 = 26$ μm).

| sample no. | $C_0/L_0$ | $D_f$ | $\phi$ | $n$ | sample no. | $C_0/L_0$ | $D_f$ | $\phi$ | $n$ |
|---|---|---|---|---|---|---|---|---|---|
| 1 | 3/13 | 1.979 | 0.946 | 1 | 17 | 1/3 | 1.893 | 0.555 | 5 |
| 2 | 3/13 | 1.979 | 0.896 | 2 | 18 | 1/5 | 1.975 | 0.960 | 1 |
| 3 | 3/13 | 1.979 | 0.849 | 3 | 19 | 1/5 | 1.975 | 0.922 | 2 |
| 4 | 5/13 | 1.938 | 0.852 | 1 | 20 | 1/5 | 1.975 | 0.885 | 3 |
| 5 | 5/13 | 1.938 | 0.726 | 2 | 21 | 3/5 | 1.723 | 0.640 | 1 |
| 6 | 5/13 | 1.938 | 0.618 | 3 | 22 | 3/5 | 1.723 | 0.410 | 2 |
| 7 | 7/13 | 1.866 | 0.710 | 1 | 23 | 3/5 | 1.723 | 0.262 | 3 |
| 8 | 7/13 | 1.866 | 0.504 | 2 | 24 | 1/7 | 1.989 | 0.980 | 1 |
| 9 | 7/13 | 1.866 | 0.358 | 3 | 25 | 1/7 | 1.989 | 0.959 | 2 |
| 10 | 9/13 | 1.746 | 0.521 | 1 | 26 | 1/7 | 1.989 | 0.940 | 3 |
| 11 | 9/13 | 1.746 | 0.271 | 2 | 27 | 3/7 | 1.896 | 0.816 | 1 |
| 12 | 9/13 | 1.746 | 0.141 | 3 | 28 | 3/7 | 1.896 | 0.666 | 2 |
| 13 | 1/3 | 1.893 | 0.889 | 1 | 29 | 3/7 | 1.896 | 0.544 | 3 |
| 14 | 1/3 | 1.893 | 0.790 | 2 | 30 | 5/7 | 1.633 | 0.490 | 1 |
| 15 | 1/3 | 1.893 | 0.702 | 3 | 31 | 5/7 | 1.633 | 0.240 | 2 |
| 16 | 1/3 | 1.893 | 0.624 | 4 | 32 | 5/7 | 1.633 | 0.118 | 3 |

32 Sierpinski carpet models with different fractal dimensions and porosity (table 2) were used to characterize the pore structure of coal samples and develop a pore-scale model for the oxygen adsorption process in coal porous media.

The adsorption and desorption reaction at the surface can be described by $A \underset{k_r}{\overset{k_f}{\rightleftharpoons}} A_s$, where $A$ and $A_s$ are, respectively, the molecular species in the gas and adsorbed on active sites, $k_f$ and $k_r$ are forward and reverse rate constants. The rate of adsorption of molecules is proportional to both the fraction of free surface sites $(1 - \theta)$ and the partial pressure of the species $(p_A)$, which can be expressed by $r_{ads} = k_f p_A (1 - \theta)$. While the rate of molecule desorption is proportional to the fraction of active sites occupied by adsorbed molecules $(\theta)$ and can be expressed with $r_{des} = k_r \theta$. To set up transport and

reaction equations in terms of bulk gas concentration ($c$) and surface concentration ($c_s$), the fraction of adsorbed molecules and partial pressure can be, respectively, substituted by [60,61]

$$\theta = \frac{c_s}{\Gamma_s} \tag{3.3}$$

and

$$p_A = cRT, \tag{3.4}$$

where $\Gamma_s$ is total concentration of the active surface of the material, $R$ is the gas constant and $T$ is the temperature. Thus, the adsorption and desorption rates can be, respectively, written as

$$r_{ads} = k_{ads}c(\Gamma_s - c_s) \tag{3.5}$$

and

$$r_{des} = k_{des}c_s, \tag{3.6}$$

where $k_{ads} = k_f RT/\Gamma_s$ and $k_{des} = k/\Gamma_s$ are the rate constant of adsorption and desorption reactions, respectively.

The material balance for the surface diffusion and reaction rate is governed by the following equation

$$\frac{\partial c_s}{\partial t} + \nabla \cdot (-D_s \nabla c_s) = r_{ads} - r_{des}, \tag{3.7}$$

where $D_s$ represents surface diffusivity. It can be also written as

$$\frac{\partial c_s}{\partial t} + \nabla \cdot (-D_s \nabla c_s) = k_{ads}c(\Gamma_s - c_s) - k_{des}c_s. \tag{3.8}$$

The convection–diffusion equation was employed to model the transport in the bulk of the porous media

$$\frac{\partial c}{\partial t} \nabla \cdot (-D\nabla c + cu) = 0, \tag{3.9}$$

where $D$ denotes the diffusivity of the reacting species, $c$ is the concentration and $u$ is the velocity.

The coupling between the mass balance in the bulk and the surface reaction is obtained as a boundary condition in the bulk's mass balance. The initial conditions for the concentration in the bulk are $c_s = 0$ and $c = c_0$. The velocity in $x$-direction equals 0 while in $y$-direction it is from the analytical expression for fully developed laminar flow between two parallel plates

$$u = \left(0, v_{max}\left[1 - \left(\frac{x - 0.5\delta}{0.5\delta}\right)^2\right]\right), \tag{3.10}$$

where $\delta$ is the plate spacing and $v_{max}$ is the local maximum speed. Insulating conditions were imposed on the surface species

$$n \cdot (-D_s \nabla c_s) = 0. \tag{3.11}$$

The boundary condition on the active surface couples the rate of the reaction on the surface with the flux of the reacting species and the concentration of the adsorbed species and bulk species

$$n \cdot (-D\nabla c + cu) = -k_{ads}c(\Gamma_s - c_s) + k_{des}c_s. \tag{3.12}$$

The boundary conditions for the inlet (bottom side) and outlet (top side) were set to be $c = c_0$ and $n \cdot (-D\nabla c + cu) = n \cdot cu$, respectively. The insulation boundary condition for the bulk problem was $n \cdot (-D\nabla c + cu) = 0$. Oxygen was selected as the adsorption gas, and the initial concentration was $9.353 \text{ mol m}^{-3}$. The dilute mass transfer module in COMSOL Multiphysics was used to solve the transient oxygen adsorption in the two-dimensional Sierpinski carpet models. The mesh was controlled by a physical mesh in which a free triangle mesh was used. The independence of the grid has been analysed and is shown in figure 3. It can be found that the meshing strategy with the maximum mesh smaller than 0.5 µm provides satisfactory solution for the example shown. In order to validate the present fractal model for gas flow in porous media, the predicted oxygen uptake was compared with available experimental data and lattice Boltzmann method (LBM) simulation. As shown in figure 4, the fractal model indicates acceptable agreement with shale adsorption data [58]

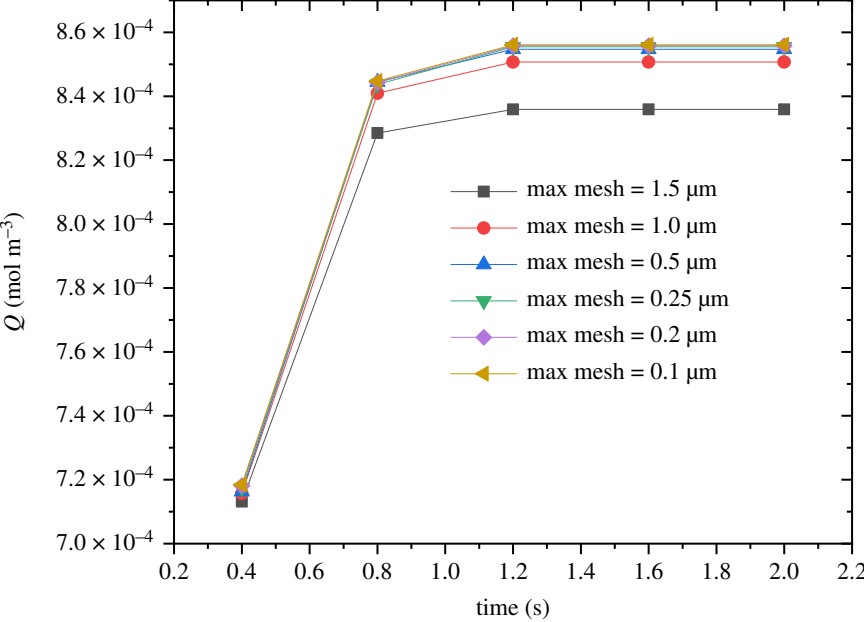

**Figure 3.** Effect of grid size on the oxygen adsorption with time.

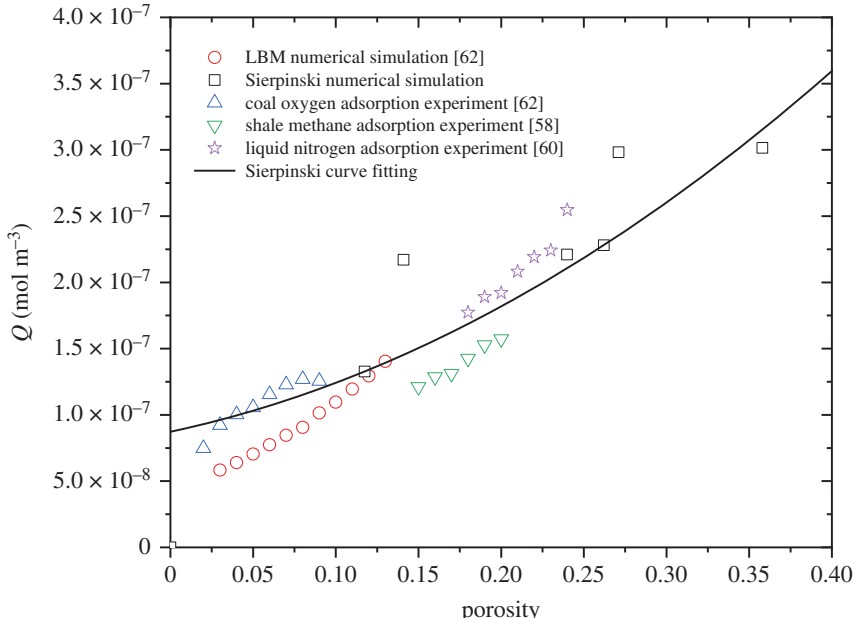

**Figure 4.** A comparison of predicted oxygen adsorption with experimental data and numerical results.

and liquid nitrogen adsorption data [60] as well as LBM results [62]. However, the predicted oxygen uptake is slightly higher than that of shale adsorption and LBM results. It may be attributed to neglecting the isolated pores in the present numerical simulation. Because of the difference of adsorption gas, the current results for oxygen uptake are smaller than that of liquid nitrogen adsorption data.

## 4. Results and discussion

The numerical results show that the rate of oxygen adsorption is very fast, it reaches steady state in nearly 2 s. It can be seen from figure 5 that the oxygen adsorption increases as the fractal dimension increases at the same time. Larger fractal dimension corresponds to higher porosity, larger pore surface area and stronger compressibility, and thus, the adsorption capacity of coal is stronger. Also, the adsorption

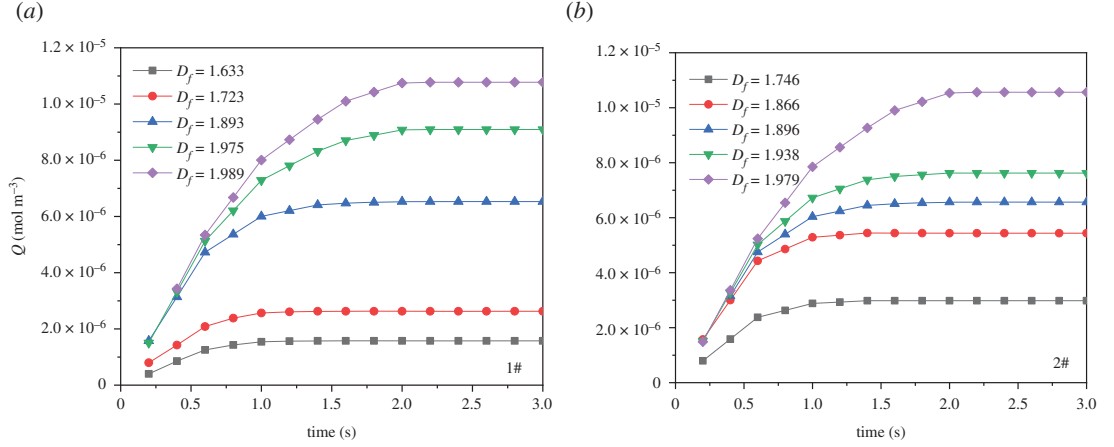

**Figure 5.** The effect of fractal dimensions on the oxygen adsorption with time.

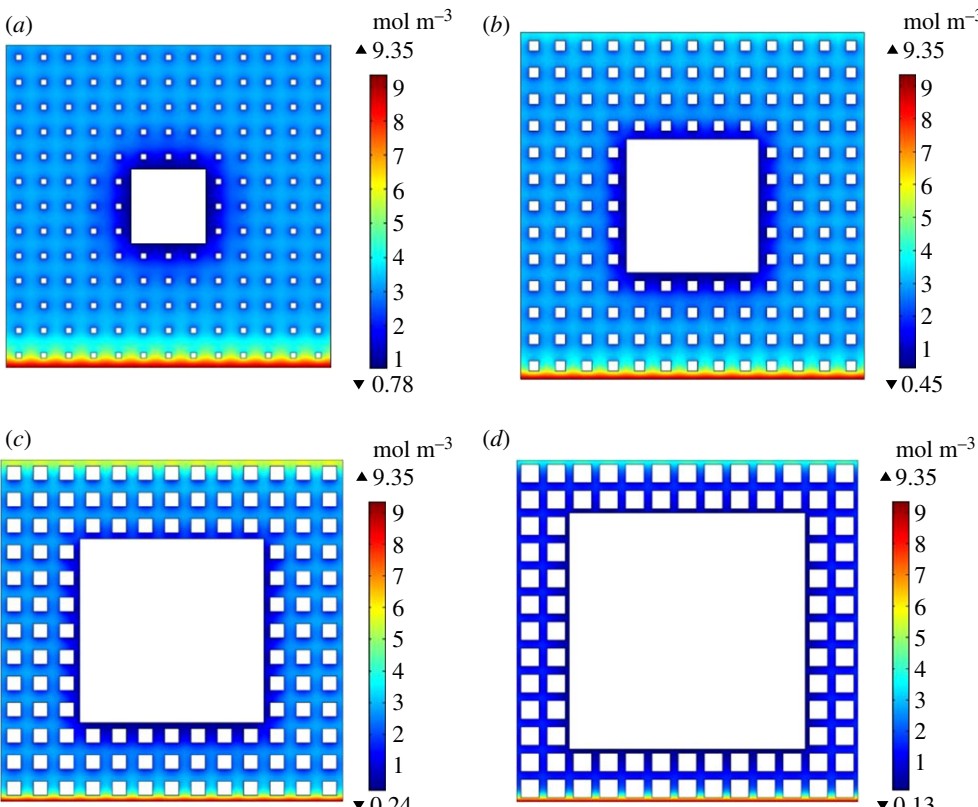

**Figure 6.** The oxygen concentration distribution in second order of the Sierpinski carpet model with $C_0/L_0$ of (a) 3/13, (b) 5/13, (c) 7/13 and (d) 9/13.

rate increases with the increment of fractal dimension. The higher the adsorption rate, the lower the permeability of coal porous media. The oxygen adsorption increases as gas concentration increases in the porous medium with low permeability. As can be seen from figure 6, the edge effect can be clearly seen near the cavity, where the concentration is lower than other locations. While the concentration near the outlet is high due to the diffusion effect. Increased iteration number leads to low porosity, and the concentration decreases. For the same order of the Sierpinski carpet model, increased cut side length ($C_0$) induces porosity and concentration to decrease.

Further calculation indicates that the oxygen adsorption increases as the porosity increases (figure 7). This is because that coal porous media with large porosity structure has loose structure and strong physical adsorption capacity. On the contrary, coal with small porosity tends to be dense and the oxygen adsorption capacity is low. According to the experimental data by Yu [62], the oxygen

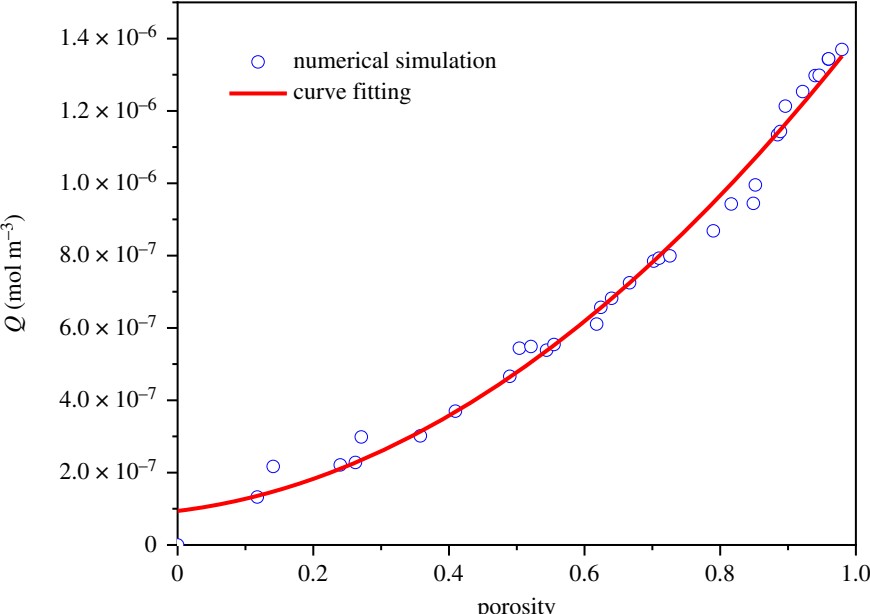

**Figure 7.** The relationship between oxygen adsorption and porosity.

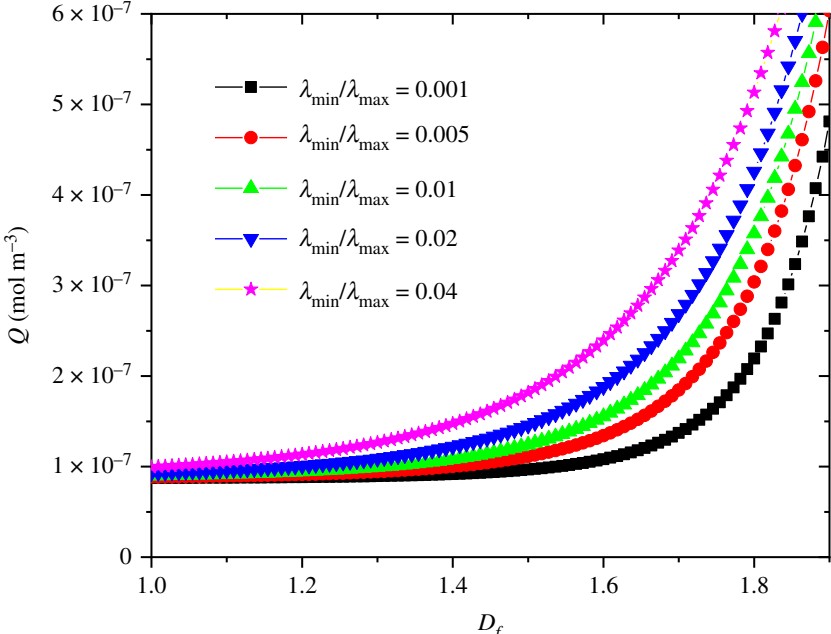

**Figure 8.** The influence of fractal dimensional and ratio of minimum to maximum pore size on the oxygen uptake.

adsorption of coal samples is negatively correlated with the temperature from 30 to 70°C, and it is positively correlated with porosity. This is consistent with the present simulation results. It can be seen from figure 7 that the relationship between oxygen adsorption and porosity can be described by $Q = 8.73 \times 10^{-8} + 2.65 \times 10^{-7}\phi + 1.04 \times 10^{-6}\phi^2$. It has been found that the oxygen adsorption depends on fractal dimension and porosity as well the ratio of minimum to maximum pore sizes, which can be related by the following formula [54,55]:

$$\phi = \left(\frac{\lambda_{\min}}{\lambda_{\max}}\right)^{d_E - D_f}, \tag{4.1}$$

where the Euclidean dimension $d_E = 2$ in a two-dimensional model. It can be seen from figure 8 that the ratio of minimum to maximum pore sizes shows important influence on the oxygen uptake. The fraction

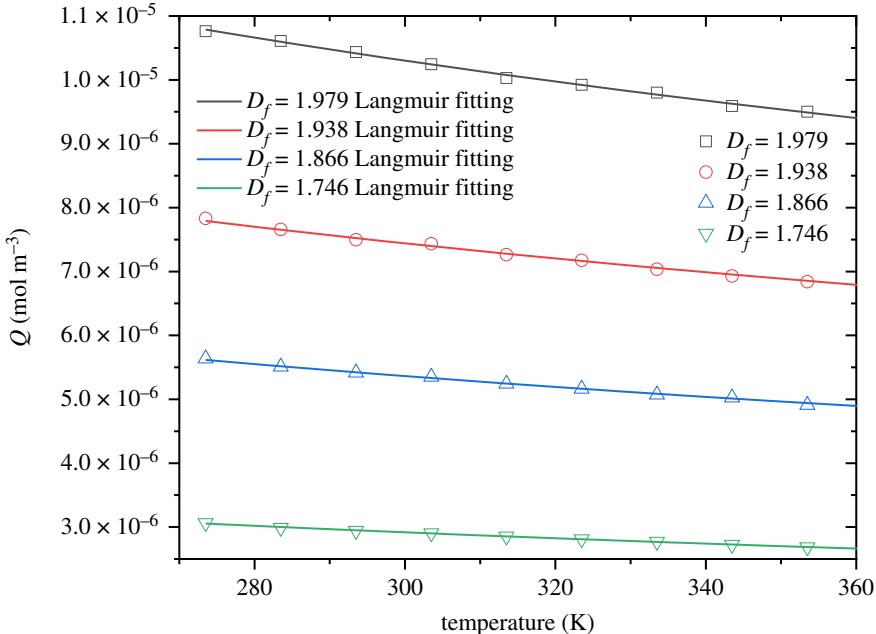

**Figure 9.** The relationship between the oxygen adsorption and temperature under different fractal dimensions.

of small pores increases as $\lambda_{min}/\lambda_{max}$ decreases under the same fractal dimension. Thus, the oxygen adsorption decreases as $\lambda_{min}/\lambda_{max}$ decreases with the same fractal dimension.

Temperature indicates significant effect on the oxygen adsorption of coal porous media. Common gas adsorption models include Langmuir equation, Freundlich equation, Temkin equation and the BET (Brunauer, Emmett and Teller) equation. The Langmuir adsorption theory is the most typical model for describing gas adsorption. According to the molecular motion theory, the amount of molecular substances adsorbed on the coal surface can be obtained

$$\mu = \frac{p}{(2\pi MRT)^{1/2}},$$
(4.2)

where $p$ is the gas equilibrium pressure, $M$ is the gas adsorption amount per unit mass of solids, $R$ is the gas constant and $T$ is temperature. The numerical results on the two-dimensional Sierpinski carpet model fit well with Langmuir equation. It is seen from figure 9 that the amount of oxygen adsorption decreases as the temperature increases. This is because the kinetic energy of oxygen molecules increases with the increase of ambient temperature, and the attraction between coal surface and oxygen hardly changes with temperature. Accordingly, oxygen is easily desorbed from the coal surface, and the amount of oxygen adsorbed by coal reduces. The temperature can activate oxygen desorption, and the higher the temperature, the more free oxygen and the less adsorbed gas [63]. Physical adsorption is an exothermic reaction, and the adsorption amount decreases as the adsorption temperature increases. That is why the adsorption process is easier to carry out at low temperature.

# 5. Conclusion

According to the liquid nitrogen adsorption experiment, the pore structures of coal samples indicate evident fractal characteristics. The fractal dimensions for low-pressure and high-pressure sections are, respectively, in the range of 1.80–2.96 and 2.45–2.82. Therefore, pore-scale model has been developed by employing the two-dimensional Sierpinski carpet model in order to study the oxygen adsorption performance of coal porous media, which agree well with available experimental data and LBM simulation. It has been found that the rate of oxygen adsorption is very fast and reaches steady state in nearly 2 s. The effect of pore structures and temperature on oxygen adsorption were discussed in detail. The oxygen adsorption increases with increased porosity, fractal dimension and the ratio of minimum to maximum pore sizes. A new relationship between oxygen adsorption and porosity $Q = 8.73 \times 10^{-8} + 2.65 \times 10^{-7}\phi + 1.04 \times 10^{-6}\phi^2$ was proposed based on present numerical results. The correlation between the oxygen adsorption and temperature was found to be consistent with

Langmuir adsorption theory. Due to the limitations of numerical simulation, it is impossible to realistically simulate the pore structure of coal and the real state of the environment. It should be noted that the present results are only valid for coal samples with statistically fractal scaling characteristics. The current work may help to understand the mechanism of coal oxygen adsorption and provides a useful reference for strategies to reduce the risk of spontaneous combustion of coal.

Data accessibility. This article does not contain any additional data.

Authors' contributions. X.Lv. carried out liquid nitrogen adsorption experiments and numerical simulation, and wrote manuscripts; X.Li. conceived the research and provided corresponding experimental equipment; P.X. established fractal model and drafted manuscript; L.C. assisted the research. All authors gave final approval for publication.

Competing interests. We declare we have no competing interests.

Funding. This work was jointly supported by the Natural Science Foundation of China (grant no. 51876196), the Zhejiang Provincial Natural Science Foundation of China (grant nos. LY18E040001 and LR19E060001) and the State Key Laboratory Cultivation Base for Gas Geology and Gas Control of Henan Polytechnic University (grant no. WS2018A02). All authors' funds come from the support of the above funds.

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
