## [Reviewer comments · Royal Society Open Science]

Review History

RSOS-191337.R0 (Original submission)

Review form: Reviewer 1

Is the manuscript scientifically sound in its present form?

Yes

Are the interpretations and conclusions justified by the results?

Yes

Is the language acceptable?

Yes

Do you have any ethical concerns with this paper?

No

Have you any concerns about statistical analyses in this paper?

No

Recommendation?

Accept with minor revision (please list in comments)

Comments to the Author(s)

Review comments

RE: RSOS-191337

Title: A Numerical Study on Oxygen Adsorption in Porous Media of Coal Rock Based on Fractal Geometry

In this manuscript, the authors studied the fractal characteristics of coal samples and developed a pore-scale model for oxygen adsorption in coal porous media by applying the self-similar fractal model. The liquid nitrogen adsorption experiments and numerical simulations were also carried out and some interesting results were found. This manuscript may be accepted for publication after minor revisions are made. Here are my comments.

1. In Introduction section, some similar works should be reviewed such as the papers by KUNJIE LI et al., *Fractals*, Vol. 26, No. 2 (2018) 1840006 and by XIANGGUO KONG et al., *Fractals*, Vol. 27, No. 5 (2019) 1950072.
2. The citation of Eq. (2) should be provided as Yu B.M. and Yao K.L., Critical percolation probabilities for site problem on Sierpinski carpets, *Z. Phys. B: Condensed Matter* 70 209-212(1988).
3. The citation of Eq. (3) should be provided as Feng Y.J. et al., A generalized model for the effective thermal conductivity of porous media based on self-similarity, *J. Phys. D: Appl. Phys.* 37, 3030-3040((2004)).
4. The citations of Eqs. (4)-(13) should be also provided before citing them.
5. If possible, the authors should provide the quantitative relationship among the oxygen uptake Q , porosity and fractal dimension D_f , and additional figures may be desirable for studying the parameter effects.

Review form: Reviewer 2**Is the manuscript scientifically sound in its present form?**

No

Are the interpretations and conclusions justified by the results?

No

Is the language acceptable?

No

Do you have any ethical concerns with this paper?

Yes

Have you any concerns about statistical analyses in this paper?

No

Recommendation?

Major revision is needed (please make suggestions in comments)

Comments to the Author(s)

Authors have tested the fractal dimension characteristics of coal in order to explain the oxygen adsorption during coal combustions. As per authors, they have carried out N_2 adsorption studied and pore modelling simulation for the purpose. The manuscript is interesting to the larger scientific community working in the field of coal combustion. However, while going through it, the following questions come to my mind.

1. Check there were instances where 'Absorption' is written, though the work is concerned with 'Adsorption'.

2. The manuscript should be thoroughly checked for grammatical correction. E.g. third sentence of the introduction is incomplete.
 3. Though low-pressure nitrogen adsorption is used to determine (D1, surface fractal dimension) and (D2, pore structure fractal dimension). Discussion part complies mostly the numerical results and doesn't cover the results of low-pressure nitrogen adsorption.
 4. Fig.4 explanation has to be made clearly.
 5. More detailed explanation needs to be provided regarding the fractal dimension determined from numerical analysis, whether it is surface or pore structure specific. If observed the Fig.8, the fractal dimension range plotted is up to 1.9, which doesn't comply with explanation provided in the pg.5, line 11 that fractal dimension values lie within the range of 2 to 3. In Figure 5 also, variation of Q is represented for a D value up to ~2.0.
 6. Main thrust area, factors affecting coal spontaneous combustion is not properly justified with the experimental and numerical results.
 7. The relationship between the oxygen uptake and porosity that is derived from numerical results have to be cross verified with experimental results.
 8. Explanation given in Figure 9, should be rethought in my suggestion, or more explanation is needed. What would happen to combustion rate in case of temperature rise and their relation with adsorption needs to be explained.
 9. Relation between oxygen intake and fractal dimension needs to be explained. In the manuscript, no experimental data is presented against the conclusion drawn from Figure 5.
 10. As explained in results & discussion part (pg 12, 158-59) increased iteration leads to low porosity. How iteration level can change the porosity of coal which is a physical property?
 11. Finally, the manuscript lacks in proper experimental data. There is a lack in proper justification between the experimental results and numerical results (fractal dimensions derived doesn't converge). Moreover, most of the results are solely drawn from numerical results even the relationship between the oxygen uptake and porosity without validating with the experimental results.
- To my view, new experimental results should be incorporated into the manuscript and may be resubmitted for further consideration.

Review form: Reviewer 3

Is the manuscript scientifically sound in its present form?

Yes

Are the interpretations and conclusions justified by the results?

Yes

Is the language acceptable?

Yes

Do you have any ethical concerns with this paper?

No

Have you any concerns about statistical analyses in this paper?

No

Recommendation?

Accept with minor revision (please list in comments)

Comments to the Author(s)

The paper presents novel concept of oxygen adsorption in porous media of coal rock based on fractal geometry. The idea is of interest and can be useful. However, there are minor issues to be addressed:

1. coal pore structure is random and has huge variability, depending on the origin of location, coal quality etc. Authors may need to comment on this and state the validity limit of this approach.
2. English text can be improved.

Decision letter (RSOS-191337.R0)

30-Oct-2019

Dear Mr Lv,

The editors assigned to your paper ("A Numerical Study on Oxygen Adsorption in Porous Media of Coal Rock Based on Fractal Geometry") have now received comments from reviewers. We would like you to revise your paper in accordance with the referee and Associate Editor suggestions which can be found below (not including confidential reports to the Editor). Please note this decision does not guarantee eventual acceptance.

Please submit a copy of your revised paper before 22-Nov-2019. Please note that the revision deadline will expire at 00.00am on this date. If we do not hear from you within this time then it will be assumed that the paper has been withdrawn. In exceptional circumstances, extensions may be possible if agreed with the Editorial Office in advance. We do not allow multiple rounds of revision so we urge you to make every effort to fully address all of the comments at this stage. If deemed necessary by the Editors, your manuscript will be sent back to one or more of the original reviewers for assessment. If the original reviewers are not available, we may invite new reviewers.

- Data accessibility

<http://datadryad.org/submit?journalID=RSOS&manu=RSOS-191337>

- Competing interests

- Authors' contributions

- Acknowledgements

- Funding statement

Kind regards,

Anita Kristiansen

Editorial Coordinator

on behalf of R. Kerry Rowe (Subject Editor)

Associate Editor's comments:

Comments to the Author:

Thank you for the manuscript submission. We're making the decision to invite a revision on the basis of the second reviewer report received. Reviewer 1 has provided a number of suggestions of citations to include; however, the Editors note these are largely their own work, and so you - as authors - should carefully consider whether these suggestions will add value to your manuscript. If they do add value, then please consider including them; however, if they do not, and may be construed as 'citation stacking', you should feel confident in not including them. Reviewer 3 observes that the quality of your written English requires attention - please visit <https://royalsociety.org/journals/authors/language-polishing/> for advice on services that you can use to assist.

Reviewer 2's comments require careful revision, and the Editors will be making their determination of how to proceed with your manuscript in response to how effectively you tackle this reviewer's comments. Please ensure you include not only a tracked changes version of the manuscript with the revision but also a full point-by-point response to the comments. Publication will be contingent on fully satisfying the reviewer that your paper is ready for publication.

Reviewers' Comments to Author:

Reviewer: 1

Comments to the Author(s)

Review comments

RE: RSOS-191337

Title: A Numerical Study on Oxygen Adsorption in Porous Media of Coal Rock Based on Fractal Geometry

In this manuscript, the authors studied the fractal characteristics of coal samples and developed a pore-scale model for oxygen adsorption in coal porous media by applying the self-similar fractal model. The liquid nitrogen adsorption experiments and numerical simulations were also carried out and some interesting results were found. This manuscript may be accepted for publication after minor revisions are made. Here are my comments.

1. In Introduction section, some similar works should be reviewed such as the papers by KUNJIE LI et al., *Fractals*, Vol. 26, No. 2 (2018) 1840006 and by XIANGGUO KONG et al., *Fractals*, Vol. 27, No. 5 (2019) 1950072.
2. The citation of Eq. (2) should be provided as Yu B.M. and Yao K.L., *Critical percolation probabilities for site problem on Sierpinski carpets*, *Z. Phys. B: Condensed Matter* 70 209-212(1988).
3. The citation of Eq. (3) should be provided as Feng Y.J. et al., *A generalized model for the effective thermal conductivity of porous media based on self-similarity*, *J. Phys. D: Appl. Phys.* 37, 3030-3040((2004)).
4. The citations of Eqs. (4)-(13) should be also provided before citing them.
5. If possible, the authors should provide the quantitative relationship among the oxygen uptake Q , porosity and fractal dimension D_f , and additional figures may be desirable for studying the parameter effects.

Reviewer: 2

Comments to the Author(s)

Authors have tested the fractal dimension characteristics of coal in order to explain the oxygen adsorption during coal combustions. As per authors, they have carried out N₂ adsorption studied and pore modelling simulation for the purpose. The manuscript is interesting to the larger scientific community working in the field of coal combustion. However, while going through it, the following questions come to my mind.

1. Check there were instances where 'Absorption' is written, though the work is concerned with 'Adsorption'.
 2. The manuscript should be thoroughly checked for grammatical correction. E.g. third sentence of the introduction is incomplete.
 3. Though low-pressure nitrogen adsorption is used to determine (D1, surface fractal dimension) and (D2, pore structure fractal dimension). Discussion part complies mostly the numerical results and doesn't cover the results of low-pressure nitrogen adsorption.
 4. Fig.4 explanation has to be made clearly.
 5. More detailed explanation needs to be provided regarding the fractal dimension determined from numerical analysis, whether it is surface or pore structure specific. If observed the Fig.8, the fractal dimension range plotted is up to 1.9, which doesn't comply with explanation provided in the pg.5, line 11 that fractal dimension values lie within the range of 2 to 3. In Figure 5 also, variation of Q is represented for a D value upto ~2.0.
 6. Main thrust area, factors affecting coal spontaneous combustion is not properly justified with the experimental and numerical results.
 7. The relationship between the oxygen uptake and porosity that is derived from numerical results have to be cross verified with experimental results.
 8. Explanation given in Figure 9, should be rethought in my suggestion, or more explanation is needed. What would happen to combustion rate in case of temperature rise and their relation with adsorption needs to be explained.
 9. Relation between oxygen intake and fractal dimension needs to be explained. In the manuscript, no experimental data is presented against the conclusion drawn from Figure 5.
 10. As explained in results & discussion part (pg 12, l 58-59) increased iteration leads to low porosity. How iteration level can change the porosity of coal which is a physical property?
 11. Finally, the manuscript lacks in proper experimental data. There is a lack in proper justification between the experimental results and numerical results (fractal dimensions derived doesn't converge). Moreover, most of the results are solely drawn from numerical results even the relationship between the oxygen uptake and porosity without validating with the experimental results.
- To my view, new experimental results should be incorporated into the manuscript and may be resubmitted for further consideration.

Reviewer: 3

Comments to the Author(s)

The paper presents novel concept of oxygen adsorption in porous media of coal rock based on fractal geometry. The idea is of interest and can be useful. However, there are minor issues to be addressed:

1. coal pore structure is random and has huge variability, depending on the origin of location, coal quality etc. Authors may need to comment on this and state the validity limit of this approach.
2. English text can be improved.

Author's Response to Decision Letter for (RSOS-191337.R0)

See Appendix A.

RSOS-191337.R1 (Revision)

Review form: Reviewer 1

Is the manuscript scientifically sound in its present form?

Yes

Are the interpretations and conclusions justified by the results?

Yes

Is the language acceptable?

Yes

Do you have any ethical concerns with this paper?

No

Have you any concerns about statistical analyses in this paper?

No

Recommendation?

Accept as is

Comments to the Author(s)

I have no more comment on this manuscript, and this manuscript can be accepted for publication.

Review form: Reviewer 3

Is the manuscript scientifically sound in its present form?

Yes

Are the interpretations and conclusions justified by the results?

Yes

Is the language acceptable?

Yes

Do you have any ethical concerns with this paper?

No

Have you any concerns about statistical analyses in this paper?

No

Recommendation?

Accept as is

Comments to the Author(s)

Authors have addressed all the comments. I recommend the paper for publication

Decision letter (RSOS-191337.R1)

17-Jan-2020

Dear Mr Lv,

It is a pleasure to accept your manuscript entitled "A Numerical Study on Oxygen Adsorption in Porous Media of Coal Rock Based on Fractal Geometry" in its current form for publication in Royal Society Open Science. The comments of the reviewer(s) who reviewed your manuscript are included at the foot of this letter.

on behalf of Prof R. Kerry Rowe (Subject Editor)
openscience@royalsociety.org

Reviewer comments to Author:
Reviewer: 1

Comments to the Author(s)
I have no more comment on this manuscript, and this manuscript can be accepted for publication.

Reviewer: 3

Comments to the Author(s)
Authors have addressed all the comments. I recommend the paper for publication

Appendix A

Response to Comments from Reviewers and Editor

Ref.: Ms. No. RSOS-191337

Title: A Numerical Study on Oxygen Adsorption in Porous Media of Coal Rock
Based on Fractal Geometry

Authors: Xianzhe Lv, Xiaoyu Liang , Peng Xu , Linya Chen

Journal: Royal Society Open Science

We have revised our manuscript according to the comments from reviewers and editor. Changes in our revised manuscript are highlighted and marked in *red color*. Here is our reply to each comment from reviewers and editor.

Reviewer #1's Comments:

In this manuscript, the authors studied the fractal characteristics of coal samples and developed a pore-scale model for oxygen adsorption in coal porous media by applying the self-similar fractal model. The liquid nitrogen adsorption experiments and numerical simulations were also carried out and some interesting results were found. This manuscript may be accepted for publication after minor revisions are made. Here are my comments.

Our Reply: We appreciate the comments by Reviewer #1 because these comments are very important for improving the quality of our paper, and we have revised our manuscript accordingly.

1. In Introduction section, some similar works should be reviewed such as the papers by KUNJIE LI et al., *Fractals*, Vol. 26, No. 2 (2018) 1840006 and by XIANGGUO KONG et al., *Fractals*, Vol. 27, No. 5 (2019) 1950072.

Our Reply: We appreciate this comment. The suggested literatures are useful and we have cited them in our revised manuscript. Please see references 39 and 40 as well as the sentences in the second paragraph on page 2.

2. The citation of Eq. (2) should be provided as Yu B.M. and Yao K.L., Critical percolation probabilities for site problem on Sierpinski carpets, *Z. Phys. B: Condensed Matter* 70 209-212(1988).

Our Reply: We are sorry about this. We have clearly indicated the reference for Eq. (2). Please see the sentence before Eq. (2) on page 3 and reference 57 in our revised manuscript.

3. The citation of Eq. (3) should be provided as Feng Y.J. et al., A generalized model for the effective thermal conductivity of porous media based on self-similarity, J. Phys. D: Appl. Phys. 37, 3030–3040((2004)).

Our Reply: We are sorry about this. We have clearly indicated the reference for Eq. (3). Please see reference 58 in our revised manuscript.

4. The citations of Eqs. (4)-(13) should be also provided before citing them.

Our Reply: We appreciate this comment. We have added references 61 and 62 for Eqs. (4)-(13).

5. If possible, the authors should provide the quantitative relationship among the oxygen uptake Q , porosity and fractal dimension D_f , and additional figures may be desirable for studying the parameter effects.

Our Reply: We appreciate this comment. The pore structure which can be characterized by porosity and fractal dimension has important influence on oxygen uptake. And the quantitative relationship between the oxygen uptake and fractal dimension is useful for studying coal spontaneous combustion. According to the present numerical results, a quantitative relationship between oxygen uptake and porosity was obtained as $Q = 8.73 \times 10^{-8} + 2.65 \times 10^{-7} \phi + 1.04 \times 10^{-6} \phi^2$. With the aid of formula between porosity and fractal dimension (Eq. (14)), the correlation between oxygen uptake and fractal dimension can be accordingly gotten. Please see the sentences before and after Eq. (14) on page 5.

Reviewer #2's Comments:

Authors have tested the fractal dimension characteristics of coal in order to explain the oxygen adsorption during coal combustions. As per authors, they have carried out N₂ adsorption studied and pore modelling simulation for the purpose. The manuscript is interesting to the larger scientific community working in the field

of coal combustion. However, while going through it, the following questions come to my mind.

Our Reply: We are very grateful to reviewer #2 for the useful comments.. We have revised our manuscript according to the review comments carefully.

1. Check there were instances where ‘Absorption’ is written, though the work is concerned with ‘Adsorption’.

Our Reply: We are very sorry about this error. We have checked our manuscript carefully and thoroughly, and correct the words.

2. The manuscript should be thoroughly checked for grammatical correction. E.g. third sentence of the introduction is incomplete.

Our Reply: We appreciate this comment and we are very sorry for the incorrect expressions. We have checked and revised our manuscript thoroughly as suggested by the reviewer.

3. Though low-pressure nitrogen adsorption is used to determine (D_1 , surface fractal dimension) and (D_2 , pore structure fractal dimension). Discussion part complies mostly the numerical results and doesn't convert the results of low-pressure nitrogen adsorption.

Our Reply: We agree with this comment. The surface fractal dimension D_1 plays a crucial role in the adsorption of coal oxygen. However, the purpose of the present work is to examine the effect of pore structures (pore size distribution) on oxygen adsorption by numerical simulation. The influence of rough pore surface on adsorption will be included in our future work.

4. Fig.4 explanation has to be made clearly.

Our Reply: We appreciate this comment. The present fractal model shows acceptable agreement with shale adsorption data and liquid nitrogen adsorption data as well as LBM results. For the slightly difference between them, we have added detailed explanation in our revised manuscript. Please see the last paragraph of section 3 on page 4.

5. More detailed explanation needs to be provided regarding the fractal dimension determined from numerical analysis, whether it is surface or pore structure specific. If observed the Fig.8, the fractal dimension range plotted is up to 1.9, which doesn't comply with explanation provided in the pg.5, line 11 that fractal dimension values lie within the range of 2 to 3. In Figure 5 also, variation of Q is represented for a D value upto~2.0.

Our Reply: We appreciate this comment. If the area porosity is assumed to be the same as volume porosity, the pore fractal dimension in two dimensional space equals to the pore fractal dimension in three dimensional space minus one. Thus, the pore fractal dimension in 2D space is in the range of 1.0 to 2.0. In order to make this clear, we have added detailed explanation about the pore fractal dimension at the end of section 2 on page 3.

6. Main thrust area, factors affecting coal spontaneous combustion is not properly justified with the experimental and numerical results.

Our Reply: We appreciate this comment. Spontaneous combustion depends on number of internal (intrinsic) and external (extrinsic) factors such as rank of coal, presence of iron pyrites, content of volatile matter, moisture content, friability/breakability, particle size and surface area of coal, ash content, atmospheric conditions, storing methods and conditions etc. The present work focused on the effect of pore structures on oxygen uptake. The other factors will be taken into account in our future work.

7. The relationship between the oxygen uptake and porosity that is derived from numerical results have to be cross verified with experimental results.

Our Reply: The relationship between oxygen uptake and porosity has been compared with experimental data and LBM numerical results in figure 4. The present fractal model indicates acceptable agreement with available experimental data and numerical results.

8. Explanation given in Figure 9, should be rethought in my suggestion, or more explanation is needed. What would happen to combustion rate in case of temperature rise and their relation with adsorption needs to be explained.

Our Reply: In order to make this clear, we have revised our explanation about Fig. 9.

Please see the last paragraph of section 4 on page 5. Generally, the ignition point of coal spontaneous combustion is around 300-350 °C. And the temperature in the present work is much lower than the ignition point of coal spontaneous combustion.

9. Relation between oxygen intake and fractal dimension needs to be explained. In the manuscript, no experimental data is presented against the conclusion drawn from Figure 5.

Our Reply: We appreciate this comment. According to the present numerical results, a quantitative relationship between oxygen uptake and porosity was obtained. With the aid of formula between porosity and fractal dimension (Eq. (14)), the correlation between oxygen uptake and fractal dimension can be accordingly gotten. Since the present fractal model has been validated by comparison with available experimental data and numerical results (shown in Fig. 4), the results shown in Fig. 5 are reliable.

10. As explained in results & discussion part (pg 12, 1 58-59) increased iteration leads to low porosity. How iteration level can change the porosity of coal which is a physical property?

Our Reply: The iteration level only makes sense for Sierpinski carpet model. The relationship between iteration level and porosity is illustrated in Eq. (3). The present work attends to generate porous structures with different porosity by adjusting the iteration level of Sierpinski carpet model.

11. Finally, the manuscript lacks in proper experimental data. There is a lack in proper justification between the experimental results and numerical results (fractal dimensions derived doesn't converge). Moreover, most of the results are solely drawn from numerical results even the relationship between the oxygen uptake and porosity without validating with the experimental results.

Our Reply: The present work aims to study the influence of pore structures on oxygen uptake with mathematical model based on fractal geometry. The fractal scaling characteristics of coal samples have been measured by liquid nitrogen adsorption experiment, which verify the present fractal model for pore structures of porous media. Based on the fractal model for pore structures, a mathematical model was developed for oxygen adsorption by finite element method (FEM). The

employed dilute mass transfer module of COMSOL Multiphysics has been extensively proven to be an effective tool for gas adsorption process. And the predicted oxygen uptake by the present model indicates good agreement with available experimental data and LBM simulation results. Therefore, the current results from the present numerical results are reliable and convincing.

Reviewer #3's Comments:

The paper presents novel concept of oxygen adsorption in porous media of coal rock based on fractal geometry. The idea is of interest and can be useful. However, there are minor issues to be addressed:

Our Reply: We appreciate the comments by Reviewer #3. We have revised the manuscript in a comprehensive and conscientious manner based on our opinions.

1. Coal pore structure is random and has huge variability, depending on the origin of location, coal quality etc. Authors may need to comment on this and state the validity limit of this approach.

Our Reply: We agree with this comment. We have clearly indicated the limitations of the present fractal model in the conclusion part. Please see the first paragraph on page 5.

2. English text can be improved.

Our Reply: We appreciate this comment. We have checked and revised our manuscript thoroughly and carefully. We hope the writing English can meet the journal's requirements.

We would like to take this opportunity to again express our great appreciation to the valuable and useful comments and suggestions by the reviewers and editor. We hope we have revised our manuscript in a satisfactory way and it can be considered for possible publication in Royal Society Open Science.

Yours sincerely,

Liang Xiaoyu

Ph.D., Associate Professor

College of Metrology & Measurement Engineering

China Jiliang University

Hangzhou, 310018, P.R. China

Email: xyliang@cjl.u.edu.cn

Peng Xu

Ph.D., Professor

College of Science, China Jiliang University

Hangzhou 310018, P.R. China

Email: xupeng@cjl.u.edu.cn